# Systemic Photoprotection in Skin Cancer Prevention: Knowledge among Dermatologists

**DOI:** 10.3390/biom11020332

**Published:** 2021-02-23

**Authors:** Luca Fania, Francesca Sampogna, Francesco Ricci, Mariafrancesca Hyeraci, Andrea Paradisi, Enzo Palese, Giovanni Di Lella, Sabatino Pallotta, Annarita Panebianco, Eleonora Candi, Elena Dellambra, Damiano Abeni

**Affiliations:** 1IDI-IRCCS, Dermatological Research Hospital, Via dei Monti di Creta 104, 00167 Rome, Italy; fg.sampogna@gmail.com (F.S.); fraric1984@gmail.com (F.R.); aparad78@gmail.com (A.P.); e.palese@idi.it (E.P.); g.dilella@idi.it (G.D.L.); s.pallotta@idi.it (S.P.); a.panebianco@idi.it (A.P.); candi@uniroma2.it (E.C.); e.dellambra@idi.it (E.D.); d.abeni@idi.it (D.A.); 2Department of Pharmaceutical and Pharmacological Sciences, University of Padua, 35121 Padua, Italy; mariafrancesca.hyeraci@phd.unipd.it; 3Dermatology Unit, “Cristo Re” General Hospital, 00167 Rome, Italy; 4Department of Experimental Medicine, University of Rome Tor Vergata, Via Montpellier, 1, 00133 Rome, Italy

**Keywords:** photoprotection, sunscreen, skin tumor, skin cancer, melanoma, nicotinamide, dermatology

## Abstract

Background: Systemic photoprotection (i.e., administration of substances such as nicotinamide, carotenoids, and vitamin D) may be important to reduce photocarcinogenesis or to support long-term protection against UV irradiation. Clinical trials showed that oral nicotinamide is effective in reducing the onset of new nonmelanoma skin cancers (NMSCs), while other oral photoprotectors failed to achieve the reduction of new melanoma or NMSC formation in humans. The aim of this study was to summarize the current state of knowledge of systemic photoprotection and to evaluate the knowledge and attitude of dermatologists regarding these treatments. Methods: The survey was conducted on a sample of dermatologists recruited according to a snowball sampling procedure. The questionnaire consisted of a first part asking for characteristics of the participant and a second part with 12 specific questions on their knowledge about systemic photoprotection, particularly their knowledge of astaxanthin, β-carotene, nicotinamide, and vitamin D3. Results: One hundred eight dermatologists answered the survey. Most of them (85.2%) stated that oral photoprotectors have a role in the prevention of skin cancer, and responses mainly mentioned nicotinamide. More than half of them (54.6%) had prescribed all the considered oral photoprotectors, but the majority of them had prescribed nicotinamide, mainly for 2 to 3 months during summer, almost invariably (*n* = 106) associated with topical photoprotectors. Most dermatologists (>80%) were aware of scientific publications demonstrating an effect of systemic photoprotectors on NMSC. Conclusions: Most Italian dermatologists have positive views on oral photoprotection in skin cancer and are aware of the demonstrated potential of nicotinamide in the prevention of NMSCs.

## 1. Introduction

Topical and systemic photoprotection approaches are the first-line prevention strategies for skin cancers, including melanoma and nonmelanoma skin cancers (NMSCs). Topical photoprotection may present limitations owing to the inadequate application or the especially short half-life on the skin, which requires frequent reapplication and risks potential side effects [1,2,3]. On the other hand, oral photoprotectors do not directly protect the skin against the damage induced by UV irradiation and may cause potential side effects, and internal factors may modify some oral photoprotector molecules. However, oral photoprotectors have some advantages, such as the ease of use, the efficiency that is not modified by external factors, and the possibility to estimate the half-lives of the different molecules [3]. Systemic photoprotection, consisting in the administration of substances such as nicotinamide, carotenoids, polyphenols, and other antioxidants, is important for reducing photocarcinogenesis or to support long-term protection against UV irradiation [3,4,5,6,7].

Notably, it has been demonstrated in clinical trials that the oral consumption of nicotinamide is effective in reducing the onset of new NMSCs [6,7,8]. Carotenoids can suppress in vitro and in vivo the formation of UVA- and UVB-mediated reactive oxygen species, preventing the photoinactivation of antioxidant enzymes and the induction of DNA damage. A randomized controlled trial studied β-carotene supplementation in the prevention of NMSC, reporting that there was no beneficial or harmful effect on the rates of new skin cancers [9]. So far, the efficacy of carotenoids in reducing new skin cancers in humans has not been demonstrated [5,9,10]. The active derivative of vitamin D (1,25(OH)_2_ D3) enhances the survival of skin cells following exposure to UV radiation by reducing the level of damage to DNA and thus reducing UV-induced apoptosis in preclinical studies [11]. Furthermore, 1,25(OH)_2_ D3 has been shown to reduce postirradiation edema, inflammation, and photocarcinogenesis in mouse skin [11]. In a recent systematic review and dose–response meta-analysis of prospective studies regarding vitamin D intake and skin cancer risk [12], the authors reported that, while intakes of dietary or supplemental vitamin D were not associated with the risk of melanoma and squamous cell carcinoma, high intakes of vitamin D from diet and supplements were slightly associated with basal cell carcinoma risk. Considering that vitamin D levels are mainly affected by exposure to the sun, a higher risk of skin cancer may be confounded by sun exposure [12]. Other antioxidants, such as astaxanthin and other molecules, fail to demonstrate the capacity to reduce new melanoma or NMSC formation [1,3].

Considering the increasing incidence of both melanoma and NMSC worldwide and the important public health issues posed by these tumors, topical and systemic photoprotection are vital to reduce the occurrence of such tumors. In fact, several factors such as the continuous increase in life expectancy in the general population, higher phototypes, and lower latitudes favor the onset of these cancers; however, the most important factor is the cumulative exposure to UV radiation.

The aim of this study was to summarize the current state of knowledge of systemic photoprotection and to evaluate the knowledge and attitude of dermatologists regarding these treatments.

## 2. Materials and Methods

The survey was conducted on a sample of dermatologists recruited by other dermatologists from among their acquaintances according to a snowball sampling procedure. The study was approved by the Institutional Ethical Committee (Approval # 608-1) of IDI-IRCCS in Rome, Italy, and was conducted in accordance with the Declaration of Helsinki. A questionnaire was sent by email to all clinicians, describing the purpose of the study. Those who agreed to participate signed a written informed consent before entering the study. Data were collected in June 2020.

The questionnaire consisted of a first part asking for characteristics of the participant, i.e., gender, number of years since they finished dermatology training (<10, 10–19, ≥20), geographical area (Northern, Central, or Southern Italy), workplace (hospital, university or research hospital, local health department, private practice). Regions included in each geographical area were as follows: in Northern Italy, Valle d’Aosta, Piemonte, Lombardia, Liguria, Veneto, Trentino Alto Adige, Friuli Venezia Giulia, and Emilia Romagna; in Central Italy, Toscana, Lazio, Umbria, and Marche; and in Southern Italy, Abruzzo, Molise, Campania, Basilicata, Puglia, Calabria, Sicilia, and Sardegna. In the second part, clinicians were asked to answer 12 specific questions about systemic photoprotection. The systemic photoprotectors which were listed in the questions were astaxanthin, β-carotene, nicotinamide, and vitamin D3.

Data were described as numbers and percentages. Results were compared in different subgroups of participants, according to gender, years since finishing dermatology training, geographical area, and workplace, using the chi-square test. Data were analyzed using IBM SPSS Statistics for Windows, Release 26.0.0.1 (IBM Corp., Armonk, NY, USA).

Two multivariable logistic models were tested using Question 2 (“Do you believe that oral photoprotectors may have a role in the prevention of skin cancers?”) and Question 7 (“Are you aware of scientific studies/trials that have demonstrated the reduction of NMSC due to one of the products indicated in systemic photoprotection?”) as dependent variables, respectively. Independent variables were gender, years since finishing dermatology training, geographical area, and workplace.

Furthermore, we reported the grade of recommendation and level of evidence regarding the efficacy of the mentioned oral photoprotectors based on clinical and preclinical studies from scientific literature in reducing the incidence of NMSC. We considered the grade of recommendation (from A to D) and the level of evidence (1a, 1b, 2a, 2b, 3a, 3b, 4, 5) based on the Centre for Evidence-Based Medicine.

## 3. Results

### Knowledge of Oral Photoprotection among Dermatologists and Grade of Recommendation or Level of Evidence of Oral Photoprotectors

One hundred eight dermatologists answered the survey. There were 55 women (51.4%). The description of the study population is reported in Table 1. Concerning Question 1 (“Do you know any supplement or drug indicated in systemic photoprotection?”) only one participant gave a negative answer. Most of them (85.2%) believed that oral photoprotectors might have a role in the prevention of skin cancer (Question 2, Table 1), especially nicotinamide (34.3%) (Question 3, Table 1). Among participants who had finished training more recently, the percentage of those who believed in the role of oral photoprotectors in skin cancer prevention was significantly higher than in the other groups. More than half of dermatologists stated that they had prescribed all the drugs listed in the questionnaire (Question 4, Figure 1). Among those who had prescribed only one drug, the majority (31.5%) had prescribed nicotinamide. According to the responses to Question 5 (“How long do you usually prescribe photoprotectors?”), most of the dermatologists prescribed systemic photoprotectors for 2 to 3 months during summer (56.1%) or for 4 to 6 months (31.8%). Almost all of them (n = 106) associated such treatment with topical photoprotectors (Question 6: “Do you usually associate them with topical photoprotectors?”). More than 80% of dermatologists were aware of scientific studies which demonstrated an effect of systemic photoprotectors on NMSC (Question 7, Table 2). The proportions were very similar to those of Question 2 about the role of photoprotectors (Question 8, Table 2). Among all dermatologists, 68.5% thought that nicotinamide was indicated in national guidelines for NMSC (Figure 2, Question 9). Only 26.9% of participants reported being aware of scientific studies that had demonstrated the reduction of melanoma due to one of the products indicated in systemic photoprotection (Table 3, Question 10). Among them, 37.9% reported that the studies concerned vitamin D3, 17.2% β-carotene, 17.2% nicotinamide, 24.1% all listed drugs, and 3.4% none of them (Answer to Question 11: “If you answered “yes” to Question 10, which photoprotector was used?”). When asking which photoprotectors were indicated in guidelines for melanoma (Question 12), 44% answered vitamin D3, 17.2% nicotinamide, 3.4% β-carotene, 13.8% all drugs, and 20.7% none of them (results not shown).

In the regression logistic model with Question 2 as the dependent variable, no significant association was found with any of the independent variables (results not shown). In Table 4 we report the results of the logistic regression model with Question 7 as the dependent variable. The association with the number of years since the end of dermatology training was significant. This means that dermatologists who had finished their training more recently (and in particular between 10 and 19 years) were more aware of studies on systemic photoprotectors for NMSC.

In Table 5 we report that only nicotinamide had an “A” grade of recommendation and a “1b” level of evidence considering clinical and preclinical studies from scientific literature in reducing the incidence of NMSC. Otherwise, β-carotene had a “D” grade of recommendation and a “1b” level of evidence. In contrast, vitamin D3 and astaxanthin presented a “D” grade of recommendation and a “5” level of evidence.

## 4. Discussion

In the present study, we observed that most of the included dermatologists considered that oral photoprotection, mainly nicotinamide, could have a role in the prevention of NMSC. Furthermore, more than half of them prescribed oral photoprotectors, again, mainly nicotinamide. Among participants, dermatologists who more recently finished training were more aware of the existence and the utilization of oral photoprotectors compared to older dermatologists. This likely depends on the fact that these treatments, and the relevant scientific studies, are quite recent. We also observed that dermatologists from Central and Northern Italy believed more in the role of oral photoprotection in the prevention of skin tumors than those from Southern Italy, while dermatologists working in hospitals believed less in photoprotection compared to dermatologists working in universities or in private practice. These data are probably also dependent on the age of participants. In fact, in the group from Southern Italy or working in hospitals we saw a higher percentage of dermatologists who had finished training in the past 20 years compared to the other areas and to the other workplaces. The multivariable model showed that dermatologists who had finished their training more recently were more aware of studies on the role of oral photoprotectors in NMSC than the older ones, taking into account gender, workplace, and geographical area. More than half of the dermatologists reported having prescribed all the drugs listed in the questionnaire, i.e., astaxanthin, β-carotene, vitamin D, and nicotinamide; however, among those who had prescribed only one drug, the majority had prescribed nicotinamide. Regarding the efficacy of systemic photoprotection in skin cancer prevention, many preclinical and clinical studies of different molecules have been reported, but only nicotinamide was demonstrated to play a role in reducing NMSC in humans [6,7,8]. Otherwise, the role in skin cancer prevention has been reported for the other molecules, mostly in preclinical studies, except β-carotene [9,11].

In our study, more than half of dermatologists prescribed systemic photoprotectors during summer, and almost one-third prescribed them for 4 to 6 months. A phase-three randomized controlled trial, carried out on 386 Australians, showed that oral nicotinamide treatment for 12 months (500 mg twice daily) was safe and effective in reducing the rates of new NMSC [7]. However, there is no evidence of benefit after nicotinamide is discontinued. It is thus very important to associate systemic photoprotection with topical photoprotectors, as was reported by almost all dermatologists in our study. Moreover, most dermatologists (81.5%) knew scientific studies which demonstrated an effect of systemic photoprotectors in NMSC prevention, mainly participants who had finished training more recently compared to the other groups. More than two-thirds of participants correctly answered that nicotinamide was the only product indicated in systemic photoprotection which had demonstrated a reduction of NMSC. Mainly dermatologists in the university and research hospitals and those who had finished training more recently answered correctly. It must be noted that considerable evidence indicates that oral nicotinamide is a photoprotective agent [13]. Nicotinamide prevents UV-induced ATP depletion, enhances UV-induced DNA repair, and mitigates the inflammation induced by environmental stressors in human keratinocytes [14]. Moreover, nicotinamide has photoprotective effects against carcinogenesis and immunosuppression in mice [15]. In human studies, a photoimmunoprotective role has been demonstrated for topical and oral nicotinamide administration. Indeed, it reduces the number of actinic keratoses [6] and protects against ultraviolet radiation-induced immunosuppression [16].

Furthermore, more than two-thirds of participants thought that nicotinamide was indicated in NMSC guidelines, confirming correct information. Specifically, nicotinamide is indicated as chemoprevention in the recent European guidelines on cutaneous squamous cell carcinoma [17,18]. On the other hand, it is not mentioned in the latest European guidelines on basal cell carcinoma, while it is mentioned in others [19,20,21]. Otherwise, about a quarter of participants confirmed they are aware of scientific studies reporting a reduction of melanoma due to systemic photoprotection, mainly vitamin D, and that this supplement was even indicated in guidelines for melanoma. A very recent study reported that nicotinamide shows a relevant antimelanoma activity in vitro and in vivo in mice, demonstrating that this molecule significantly delayed tumor growth in vivo and improved survival of melanoma-bearing mice [22]. However, no studies demonstrated that oral photoprotection has a role in melanoma prevention in humans, and no oral photoprotector is indicated in guidelines for melanoma.

The limitations of this study included the limited number of enrolled dermatologists, the necessary use of self-reported measures, and the setting circumscribed to the Italian population.

## 5. Conclusions

We found that most Italian dermatologists believed in the role of oral photoprotection in skin cancer prevention and more than half of them prescribed it. They were aware of the role of nicotinamide in the prevention of NMSCs, but there were still some doubts about the knowledge of oral photoprotection in melanoma prevention. Considering the importance of skin cancers, which present a progressive increase in incidence and an impressive cost of treatment, it is crucial to improve the knowledge of systemic photoprotection among dermatologists.

## Figures and Tables

**Figure 1 biomolecules-11-00332-f001:**
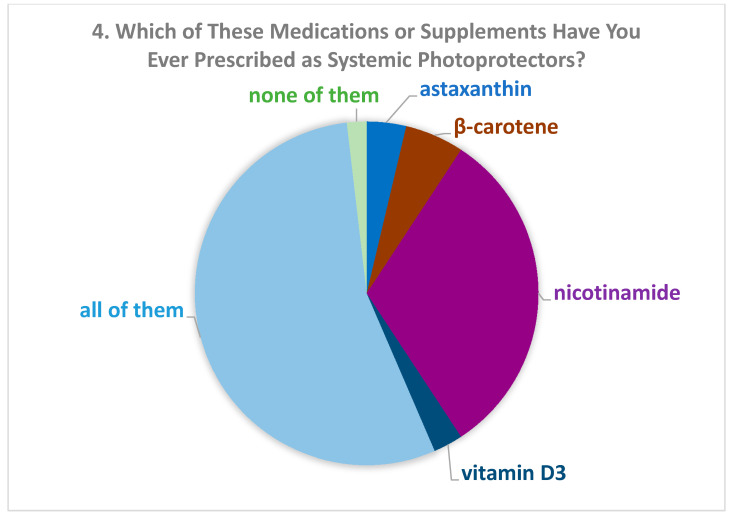
Percentage of drugs or supplements prescribed by the dermatologists.

**Figure 2 biomolecules-11-00332-f002:**
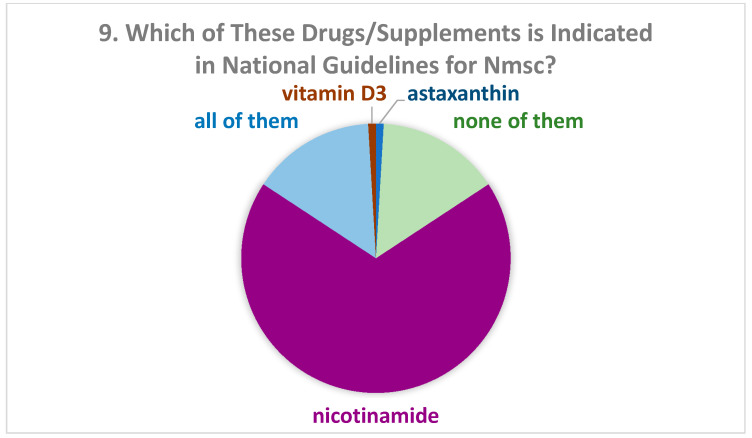
Percentage of drugs or supplements which dermatologists believed to be indicated in national guidelines for NMSC.

**Table 1 biomolecules-11-00332-t001:** Description of the characteristics of the dermatologists participating in the study and of their answers to the questions concerning their beliefs about oral photoprotectors.

			2. Do You Believe That These Drugs or Supplements (Oral Photoprotectors) May Have a Role in the Prevention of Skin Cancers?	3. If So, Which of These Drugs or Supplements Do You Believe May Play a Role in Systemic Photoprotection? Possible Answers: Astaxanthin, Β-Carotene, Nicotinamide, Vitamin D3, All of Them, None of Them.				
			YES	β-Carotene	Nicotinamide	Vitamin D3	All of Them	None of Them
		N (column%)	N (row%)	N (row%)	N (row%)	N (row%)	N (row%)	N (row%)
Overall		108 (100)	92 (85.2)	2 (1.9)	37 (34.3)	3 (2.8)	63 (58.3)	3 (2.8)
Gender	Male	52 (48.1)	43 (82.7)	0	19 (36.5)	1 (1.9)	29 (55.8)	3 (5.8)
	Female	56 (51.9)	49 (87.5)	2 (3.6)	18 (32.1)	2 (3.6)	34 (60.7)	0
Years since end of training	<10	33 (30.6)	32 (97.0)	2 (6.1)	10 (30.3)	1 (3.0)	20 (60.6)	0
	10–19	28 (25.9)	26 (92.9)	0	13 (46.4)	2 (7.1)	13 (46.4)	0
	≥20	47 (43.5)	34 (72.3) *	0	14 (29.8)	0 (0.0)	30 (63.8)	3 (6.4)
Area	Northern	24 (22.2)	21 (87.5)	1 (4.2)	8 (33.3)	3 (12.5)	12 (50.0)	0
	Central	61 (56.5)	55 (90.2)	0	22 (36.1)	0 (0.0)	38 (62.3)	1 (1.6)
	Southern	23 (21.3)	16 (69.6) **	1 (4.3)	7 (30.4)	0 (0.0)	13 (56.5)	2 (8.7)
Workplace	Hospital	17 (15.7)	11 (64.7) ***	0	5 (29.4)	2 (11.8)	10 (58.8)	0
	University/research hospital	47 (43.5)	42 (89.4)	0	19 (40.4)	0	26 (55.3)	2 (4.3)
	Local health department	15 (13.9)	14 (93.3)	1 (6.7)	6 (40.0)	1 (6.7)	7 (46.7)	0
	Private practice	29 (26.9)	25 (86.2)	1 (3.4)	7 (24.1)	0	20 (69.0)	1 (3.4)

* *p* < 0.001 from chi-square test compared with the categories “<10” and “10–19” grouped together. ** *p* < 0.05 from chi-square test compared with the categories “Northern” and “Central” grouped together. *** *p* < 0.01 from chi-square test compared with the other categories grouped together.

**Table 2 biomolecules-11-00332-t002:** Description of the answers of dermatologists to the questions concerning their awareness of studies on the role of oral photoprotectors in nonmelanoma skin cancer (NMSC).

		7. Are You Aware of Scientific Studies/Trials That Have Demonstrated the Reduction of NMSC Due to One of the Products Indicated in Systemic Photoprotection?	8. If So, Which of These Drugs or Supplements? Possible Answers: Astaxanthin, β-Carotene, Nicotinamide, Vitamin D, All of Them, None of Them.				
		Yes	Nicotinamide	Astaxanthin	Vitamin D3	All of Them	None of Them
		N (row%)	N (row%)	N (row%)	N (row%)	N (row%)	N (row%)
Overall		88 (81.5)	74 (68.5)	1 (0.9)	1 (0.9)	16 (14.8)	16 (14.8)
Gender	Male	43 (82.7)	33 (63.5)	1 (1.9)	0	9 (17.3)	9 (17.3)
	Female	45 (80.4)	38 (67.9)	0	1 (1.8)	9 (16.1)	8 (14.3)
Years since end of training	<10	30 (90.9)	28 (84.8)	0	0	3 (9.1)	2 (6.1)
	10–19	26 (92.9)	23 (82.1)	1 (3.6)	0	2 (7.1)	2 (7.1)
	≥20	32 (68.1) *	20 (42.6)	0	1 (2.1)	13 (27.7)	13 (27.7)
Area	Northern	19 (79.2)	43 (70.5)	0	0	4 (16.7)	8 (13.1)
	Central	52 (85.2)	16 (66.7)	0	1 (1.6)	9 (14.8)	4 (16.7)
	Southern	17 (73.9)	12 (52.2)	1 (4.3)	0	5 (21.7)	5 (21.7)
Workplace	Hospital	13 (76.5)	11 (67.4)	1 (5.9)	0	2 (11.8)	3 (17.6)
	University/research hospital	40 (85.1)	34 (72.3)	0	1 (1.2)	6 (12.8)	6 (12.8)
	Local health department	11 (73.3)	10 (66.7)	0	0	2 (13.3)	3 (20.0)
	Private practice	24 (82.8)	16 (55.2)	0	0	8 (27.6)	5 (17.2)

* *p* < 0.01 from chi-square test compared with the categories “<10” and “10–19” grouped together.

**Table 3 biomolecules-11-00332-t003:** Description of the answers of dermatologists to the question concerning their awareness of studies on the role of oral photoprotectors in melanoma.

		10. Are You Aware of Scientific Studies/Trials That Have Demonstrated the Reduction of Melanoma Due to One of the Products Indicated in Systemic Photoprotection?
		Yes (row%)
Overall		29 (26.9)
Gender	Male	15 (28.8)
	Female	14 (25.0)
Years since end of training	<10	12 (36.4)
	10–19	8 (28.6)
	≥20	9 (19.1)
Area	Northern	10 (41.7)
	Central	12 (19.7)
	Southern	7 (30.4)
Workplace	Hospital	7 (41.2)
	University/research hospital	12 (25.5)
	Local health department	2 (13.3)
	Private practice	8 (27.6)

**Table 4 biomolecules-11-00332-t004:** Results of the multivariable logistic regression model with Question 7 (“Are you aware of scientific studies/trials that have demonstrated the reduction of NMSC due to one of the products indicated in systemic photoprotection?”) as the dependent variable.

Model				*p* Value	Exp (B)	95.0% Confidence Interval for Exp (B)	
	Levels	B	Standard Error			Lower Limit	Upper Limit
Gender	Female vs. male	−0.716	0.606	0.237	0.489	0.149	1.601
Years since end of training	≥30 (Ref)	--					
	<10	1.426	0.771	0.064	4.16	0.918	18.854
	10–19	2.101	0.899	0.019	8.177	1.405	47.58
	20–29	−0.822	0.735	0.263	0.44	0.104	1.856
Geographical area	Southern (Ref)	--					
	Central	0.784	0.756	0.299	2.191	0.498	9.632
	Northern	0.1	0.834	0.905	1.105	0.216	5.665
Workplace	Private practice (Ref)	--					
	Hospital	−0.002	0.888	0.998	0.998	0.175	5.685
	University/research hospital	−0.084	0.709	0.905	0.919	0.229	3.687
	Local health department	−1.084	0.863	0.209	0.338	0.062	1.835

Ref = Reference category.

**Table 5 biomolecules-11-00332-t005:** Clinical and preclinical studies, grade of recommendation, and level of evidence regarding different oral photoprotectors in the prevention of nonmelanoma skin cancer.

Molecule	Studies Regarding Skin Cancer Prevention						Grade of Recommendation *	Level of Evidence *
	Preclinical Studies	Case Reports/Series	Observational Studies	Randomized Controlled Trials	Systematic Reviews/Meta-Analyses	Efficacy in Prevention of NMSC		
Nicotinamide	+	−	−	+	−	+	A	1b
β-Carotene	+	−	−	+	−	−	D	1b
Vitamin D3	+	−	+	−	+	−	D	5
Astaxanthin	+	−	−	−	−	−	D	5

* From the Centre for Evidence-Based Medicine, https://www.cebm.ox.ac.uk/resources/levels-of-evidence/oxford-centre-for-evidence-based-medicine-levels-of-evidence-march-2009 (accessed date: 30 December 2020). “+” means presence of studies while “−“ means no studies regarding that kind of research.

## Data Availability

Data are contained within the article.

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
