# Peer review of "Systemic Photoprotection in Skin Cancer Prevention: Knowledge among Dermatologists"

_biomolecules, 2021, doi:10.3390/biom11020332_

Round 1

Reviewer 1 Report

This is a nice survey study among 108 dermatologist in Italy that shows real life assumptions and prescription behavior on systemic photoprotection. The results are supplemented by an overview of studies that have evaluated the significance of systemic photoprotection.

In summary the data are clearly presented. In fact, I have no further comments

Author Response

This is a nice survey study among 108 dermatologist in Italy that shows real life assumptions and prescription behavior on systemic photoprotection. The results are supplemented by an overview of studies that have evaluated the significance of systemic photoprotection.

In summary the data are clearly presented. In fact, I have no further comments

We thank the reviewer for his/her comment.

Reviewer 2 Report

Having read the manuscript "Systemic Photoprotection in Skin Cancer Prevention: Knowledge among Dermatologists" I have the following comments:

  1.  What geographic areas define Northern, Central and Southern Italy?
  2. The survey really only asks one question is whether the dermatologist is aware of photo protection and what do they know about it.
  3. Why was no question asked about whether the dermatologists prescribe photoprotection to their patients?
  4. What evidence is there that it is having an effect on these patients?
  5. In the survey what does Q1, 5 & 6 ask.  I see Q4 and 9 is shown as a figure, which could easily be shown as a table.
  6. The tables should be in landscape format not portrait format, it is not easy to read the headings.
  7. In Table 1-3 Replace Sex with Gender, and Area subdivision should be Northern, Central and Southern. 
  8. Table 4 write β-carotene, and Vitamin D3 (not vitamin D) instead of what is written there.
  9. Ref 1, in which city was this book published?
  10. Ref 17, this citation is incomplete.  Please correct.

Author Response

Reviewer: 2

Comments to the Author

  1.  What geographic areas define Northern, Central and Southern Italy?

We thank the reviewer for his/her comment and we specified all the different Italian regions in the “Materials and Methods” section.

  1. The survey really only asks one question is whether the dermatologist is aware of photo protection and what do they know about it. Why was no question asked about whether the dermatologists prescribe photoprotection to their patients?

We thank the reviewer for his/her comment. In fact, we do have a question about whether the dermatologists prescribe photoprotection to their patients. Figure 1 reported the percentage of drugs or supplement prescribed by the dermatologists

  1. What evidence is there that it is having an effect on these patients?

This is an interesting question, however, we did not include such information in our questionnaire.

  1. In the survey what does Q1, 5 & 6 ask.  I see Q4 and 9 is shown as a figure, which could easily be shown as a table.

We thank the reviewer for his/her comment and we specified Q1, 5 and 6 in the “Results” section.

  1. The tables should be in landscape format not portrait format, it is not easy to read the headings.

We thank the reviewer for his/her comment. The tables were indeed in the landscape format, however, when we copied and pasted them in the main document, the tables changed their format. Therefore, we decided to enclose them in a separate file.

  1. In Table 1-3 Replace Sex with Gender, and Area subdivision should be Northern, Central and Southern. 

We made these corrections for all tables.

  1. Table 4 write β-carotene, and Vitamin D3 (not vitamin D) instead of what is written there.

We made these corrections for all tables and in the text.

  1. Ref 1, in which city was this book published?

We added it in the reference 1.

  1. Ref 17, this citation is incomplete.  Please correct.

We modified the reference.

Reviewer 3 Report

The paper review the level of knowledge of some molecules among Italian dermatologists. The paper could be shortened, highlighting only the main results.

One important issue is that the authors have mixed the role in photoprotection and the role as cancer chemoprevention in some molecules. All photoprotective molecules could prevent cancer, but not al chemopreventing molecules are photoprotective (eg retinoids). Nicotinamide has a possible photoprotective effect, but the role in NMSC seems to be of chemoprevention. The authors should discuss this.

The authors have only performed bivariate analysis and it seems multivariate analysis could have given them some insights as described in line 186, but this has not been performed.

Line 18. systemic photoprotection "may be" important. At this stage, there is not enough evidence to suggest that "is" important. Reducing the onset of new NMSC may not be due to a "systemic photoprotection" effect.

Line 43. Why is important "systemic efficacy"? Sunlight should be blocked and this is not a systemic effect.

Line 44. Potential side effects are also a limitation for systemic products.

Line 46. Systemic molecules will not be modified by external factors, but may be modified by internal ones.

Line 108. How many dermatologist were surveyed? What was the percentage of response?

Line 112. Instead of "were specialised", it would read better "finished Dermatology training" or "finished training". This could be used in other parts of the paper.

Line 211 highlights an important issue, nicotinamide has been published as chemoprevention but not as systemic photoprotection. The authors should clarify this.

Author Response

Reviewer: 3

Comments to the Author

The paper review the level of knowledge of some molecules among Italian dermatologists. The paper could be shortened, highlighting only the main results.

One important issue is that the authors have mixed the role in photoprotection and the role as cancer chemoprevention in some molecules. All photoprotective molecules could prevent cancer, but not al chemopreventing molecules are photoprotective (eg retinoids). Nicotinamide has a possible photoprotective effect, but the role in NMSC seems to be of chemoprevention. The authors should discuss this.

We thank the reviewer for this important comment and we added this sentence to clarify the role of nicotinamide as photoprotective agent in line 227-235.

The authors have only performed bivariate analysis and it seems multivariate analysis could have given them some insights as described in line 186, but this has not been performed.

We have now tested two multivariable models for Question 2 (“Do you believe that oral photoprotectors may have a role in the prevention of skin cancers?” and 7 (“Are you aware of scientific studies/trials that have demonstrated the reduction of NMSC due to one of the products indicated in systemic photoprotection?”), which are the questions which better summarize dermatologists’ knowledge on systemic photoprotection. Independent variables were: gender, years since the end of dermatology training, geographical area, and workplace. Since in the first model all variables were not significantly associated, we reported such results in the text and showed only the results for Question7 (Table 4).

Line 18. systemic photoprotection "may be" important. At this stage, there is not enough evidence to suggest that "is" important. Reducing the onset of new NMSC may not be due to a "systemic photoprotection" effect.

We modified the sentence

Line 43. Why is important "systemic efficacy"? Sunlight should be blocked and this is not a systemic effect.

We modified the sentence: “A lack of systemic efficacy” has been removed.

Line 44. Potential side effects are also a limitation for systemic products.

We added “potential side effects” in the sentence in line 45.

Line 46. Systemic molecules will not be modified by external factors, but may be modified by internal ones.

We modified the sentence in line 46.

Line 108. How many dermatologist were surveyed? What was the percentage of response?

We thank the reviewer for his/her comment. Since snowball sampling is a non-probability sampling technique, it is not possible to define a source population. However, participants who recruited other participants reported an acceptance rate of approximately 80%.

Line 112. Instead of "were specialised", it would read better "finished Dermatology training" or "finished training". This could be used in other parts of the paper.

We modified the sentences.

Line 211 highlights an important issue, nicotinamide has been published as chemoprevention but not as systemic photoprotection. The authors should clarify this.

We thank the reviewer for this important comment and we added this sentence to clarify the role of nicotinamide as photoprotective agent in line 227-235:

It has to be specified that considerable evidence indicate that oral nicotinamide is a photoprotective agent (Chen, 2014). Nicotinamide prevents UV-induced ATP depletion, enhances UV-induced DNA repair and mitigates the inflammation induced by environ-mental stressors in human keratinocytes (Surjana, 2013). Moreover, nicotinamide has photoprotective effects against carcinogenesis and immunosuppression in mice (Surjana, 2010). In human studies, topical and oral nicotinamide administration display a photo-immunoprotective role. Indeed, it reduces the number of actinic keratoses (Surjana, 2012) and protects against ultraviolet radiation-induced immunosuppression (Yiase-mides, 2009).

Round 2

Reviewer 2 Report

I would like to thank the authors for making the revisions as asked and have no further comments/suggestions to make.

Author Response

We thank the reviewer for his/her comment.

Reviewer 3 Report

The paper reads better after the corrections, and the multivariable analysis has shown that many of the "findings" were not significant after correction for other factors. It is a very long paper for the type of new information provided. 

Author Response

Comments and Suggestions for Authors

The paper reads better after the corrections, and the multivariable analysis has shown that many of the "findings" were not significant after correction for other factors. It is a very long paper for the type of new information provided. 

 We thank the reviewer for his/her comment. The paper has been shortened and the results improved.